# GENERATIVE DIFFUSION MODELS FOR HIGH-DIMENSIONAL TIME SERIES

## ABSTRACT

We propose a two-stage pipeline for high dimensional time series generation: (i) nonparametric kernel estimation for the conditional first and second moments of the underlying data increments to recover residuals, and (ii) score-based diffusion model trained on these residuals. We derive finite-time convergence estimates for reverse-time sampling in both total variation (TV) and Wasserstein-2 ($W_2$), with explicit dependence on the variance preserving noise schedule. Experiments on synthetic multivariate processes validate: (a) empirical TV and $W_2$ track the theoretical upper bounds, and (b) Monte Carlo estimates of test functionals achieve the predicted standard errors.

## 1 INTRODUCTION

Time-reversed diffusion models have emerged as an interesting approach to generative modeling (Sohl-Dickstein et al. (2015); Song & Ermon (2019); Ho et al. (2020); Song et al. (2021)), achieving significant empirical success in image, audio, and text synthesis, of which DALL-E and SORA are perhaps the most well-known examples. There are two main types of diffusion models: denoising diffusion probabilistic models (DDPMs) (Ho et al. (2020), Dhariwal & Nichol (2021)) and denoising diffusion implicit models (DDIMs) Song et al. (2020), in which the diffusion processes are non-Markovian. We utilize DDPMs to motivate our methodology.

DDPMs are comprised of a forward process and a reverse process. The forward *noising* process is characterized by a stochastic differential equation (SDE) initialized using the empirical distribution of a data sample. The forward distribution is often chosen to be ergodic, with a known stationary distribution, typically Gaussian. Given the forward process, we can construct a corresponding time-reversed process, called the *denoising* process. To generate samples from the target data distribution, we simulate the reverse process starting from an I.I.D. initialization with a Gaussian distribution.

**Related work.** Generative modeling for multivariate time series poses multiple challenges, particularly preserving complex temporal structure. It is not enough to learn the marginal distribution or even the joint distribution without exploiting the sequential nature of the data. We instead require a conditional generative model that generates each observation considering the past observations. Recent time-series generators have introduced more powerful techniques involving Generative Adversarial Networks (GANs) Yoon et al. (2019) and Variational Autoencoders (VAEs) Bühler et al. (2020). Diffusion models have also driven much of the progress for time series tasks such as imputation and forecasting (Rasul et al. (2021), Kollovieh et al. (2023), Yang et al. (2024), Yuan & Qiao (2024), Su et al. (2025)).

**Contributions**. We introduce an algorithm that involves a Nadaraya-Watson kernel estimator to decompose the time series into its conditional mean, covariance and residuals, followed by training a score-based diffusion model on these extracted residuals. Our convergence analysis is complimentary to recent work on (i) generalization of learned scores Stéphanovitch et al. (2025), (ii) regularity beyond log-concavity (Stéphanovitch (2025), Gentiloni-Silveri & Ocello (2025)), and (iii) explicit KL/$W_2$ for score-based generative model families Conforti et al. (2024) and noise-schedule sensitivity analysis Strasman et al. (2025). The TV and $W_2$ bounds that we provide are novel in that they make the dependence on the noise schedule explicit and decouple initialization, score, and discretization errors via a Grönwall coupling.

## 2 DESCRIPTION OF ALGORITHM

Let $X_{t_k} \in \mathbb{R}^d$ denote the observations, where $t_k = k\Delta t, \ k = 1, \ldots, N$ with $\Delta t$ timesteps. We seek to estimate the first and second conditional moments of the data:

$$\mu(x) = \lim_{\Delta t \to 0} \frac{1}{\Delta t} \mathbb{E}[\Delta X_t \mid X_t = x] \tag{1}$$

$$a(x) = \lim_{\Delta t \to 0} \frac{1}{\Delta t} \mathrm{Cov}(\Delta X_t \mid X_t = x), \tag{2}$$

where $a(x) = \sigma^\top \sigma(x) \in \mathbb{R}^{d \times d}$ is the conditional covariance matrix of the increments. To do this, we utilize the Nadaraya-Watson kernel estimator Nadaraya (1964); Watson (1964); Nadaraya (1970). The estimators are given by:

$$\widehat{\mu}(x) = \frac{\sum_{k=1}^N K_h(x - X_{t_k})\Delta X_{t_k}}{W(x)} \tag{3}$$

$$\widehat{a}(x) = \frac{\sum_{k=1}^N K_h(x - X_{t_k})(\Delta X_{t_k} - \hat{\mu}(x))(\Delta X_{t_k} - \hat{\mu}(x))^\top}{W(x)}, \tag{4}$$

where $W(x) = \Delta t \sum_{k=1}^N K_h(x - X_{t_k})$ for $K_h(x)$ kernel function with bandwidth $h$, and $\Delta X_{t_k} = X_{t_{k+1}} - X_{t_k}$. The bandwidth $h$ is chosen in a locally adaptive $k$ nearest neighbors manner. Define now $\widehat{\sigma}(x)$ as a Cholesky square root of $\widehat{a}(x)$:

$$\widehat{\sigma}^\top(x)\widehat{\sigma}(x) = \widehat{a}(x) \tag{5}$$

We may define the *residuals*

$$\widehat{\epsilon_{t_i}^{(n)}} = \widehat{\sigma}^\top(X_{t_i}^{(n)})^{-1}[\Delta X_{t_i}^{(n)} - \widehat{\mu}(X_{t_i}^{(n)})]. \tag{6}$$

**Remark 1.** *Note that, as the square root of the matrix $\widehat{a}$ is only defined up to a rotation, we cannot hope to recover a consistent estimator of $\sigma(x)$ i.e that $\widehat{\sigma}(x) \to \sigma(x)$. However, as we will see, under high-frequency asymptotics on the observed path we will typically have $\widehat{a}(x) \to a(x)$ i.e. we recover $\sigma(x)$ up to a (local) rotation. This means we cannot interpret the $\epsilon_{t_k}$ as a "filtering" of the noise terms, but these residuals allow us to recover, asymptotically, the second order structure of $\epsilon_t$.*

**Remark 2.** *Our nonparametric estimation captures temporal dependence to the extent it is included in the conditioning set. In the simplest implementation, we use the current state $X_t$ as the kernel input, which yields an effectively first–order Markov model in $X_t$. For non-Markov dynamics it is natural to augment the kernel input with lagged covariates $S_t = (X_t, X_{t-1}, \ldots, X_{t-L+1})$ for lag length $L$, and to restrict the kernel weights to past observations only by using an adaptive $k$–nearest–neighbour bandwidth.*

Once these residuals are filtered, we may feed it into the score-based diffusion model for generating new samples. We use a time dependent Ornstein-Uhlenbeck (OU) process for the forward SDE:

$$\begin{aligned} \mathrm{d}X_t &= -\tfrac{1}{2}\beta_t X_t \mathrm{d}t + \sqrt{\beta_t}dW_t \\ X_0 &\sim p_0, \end{aligned} \tag{7}$$

where $\beta_t$ is a time-dependent function. Let us define $\alpha_t = \int_0^t \beta_s \mathrm{d}s$. Then the reverse SDE is given by

$$\begin{aligned} \mathrm{d}Y_t &= \tfrac{1}{2}\beta_{T-t}Y_t\mathrm{d}t + \beta_{T-t}\nabla \log p_{T-t}(Y_t)\mathrm{d}t + \sqrt{\beta_{T-t}}\mathrm{d}W_t, \\ Y_0 &\sim \mathcal{N}(m_T x_0, v_T I), \end{aligned} \tag{8}$$

where $m_t = \exp(-\tfrac{1}{2}\alpha_t)$ and $v_t = 1 - \exp(-\alpha_t)$. Note that $X_t \stackrel{d}{=} m_t X_0 + \sqrt{v_t}\epsilon$ where $\epsilon \sim \mathcal{N}(0, I)$, so that the *exact* score function is

$$\nabla \log p_{t|0}(x \mid x_0) = \frac{m_t x_0 - x}{v_t} \stackrel{d}{=} -\frac{\epsilon}{\sqrt{v_t}}. \tag{9}$$

We define a score network $-\sqrt{v_t} \cdot s_\theta(X_t, t)$ that then predicts the noise $\epsilon$ from the noisy data $X_t \stackrel{d}{=} m_t X_0 + \sqrt{v_t}\epsilon$. Then the denoising score matching objective becomes

$$\mathbb{E}_{x_0 \sim p_{\text{data}}} \mathbb{E}_{x \sim p_{t|0}} \left[ \left\| \frac{m_t X_0 - x}{v_t} - s_\theta(X_t, t) \right\|^2 \right] = \mathbb{E}_{x_0 \sim p_{\text{data}}} \mathbb{E}_{\epsilon \sim \mathcal{N}(0, I)} \left[ \left\| s_\theta(m_t X_0 + \sqrt{v_t}\epsilon, t) + \frac{\epsilon}{\sqrt{v_t}} \right\|^2 \right].$$
(10)

See Appendix A for background on score-based diffusion models. Algorithm 1 outlines the kernel estimation, residual extraction, and score-based diffusion model training, all which occur offline. We use 9 as the conditional target for training our score network. Algorithm 2 outlines the generation of synthetic data samples.

---

**Algorithm 1** Kernel estimation, residual extraction, and score model training

---

**Input:** Observations $X_{t_k} \in \mathbb{R}^d$ with $k = 1, ..., N$ where $t_k = k\Delta t$, $\Delta X_{t_k} = X_{t_{k+1}} - X_{t_k}$.

**for** $x \in \mathbb{D} \subset \mathbb{R}^d$ **do**          ▷ Kernel Estimation

     Compute weight denominator $W(x) = \Delta t \sum_{k=1}^N K_h(x - X_{t_k})$ for $K_h(x)$ kernel function with bandwidth $h$.

     Compute $\widehat{\mu}(x) = \frac{\sum_{k=1}^N K_h(x - X_{t_k}) \Delta X_{t_k}}{W(x)}$.

     Compute $\widehat{a}(x) = \frac{\sum_{k=1}^N K_h(x - X_{t_k})(\Delta X_{t_k} - \hat{\mu}(x))(\Delta X_{t_k} - \hat{\mu}(x))^\top}{W(x)}$

     Compute $\widehat{\sigma}(x) = \texttt{CholeskySqrt}(\widehat{a}(x))$.

**end for**

**for** $k = 1$ to $N$ **do**          ▷ Residuals

     $\epsilon_{t_k} = \widehat{\sigma}^\top(X_{t_k})^{-1}[\Delta X_{t_k} - \widehat{\mu}(X_{t_k})]$

**end for**

         ▷ Offline: learning to generate the residuals

Precompute **noise schedule** $\beta_t = \beta_{\max}^{1-t} \beta_{\min}^t$, $m_t = \exp(-0.5 \int_0^t \beta_s \mathrm{d}s)$, and $v_t = 1 - m(t)^2$.

**while** current_iteration $<$ Max_iterations **do**

     Sample a minibatch $\{x_0^{(b)}, b \in B\} \subset \{\widehat{\epsilon_{t_k}}, k = 1, \ldots, N\}$, $(t^{(b)} \sim \texttt{UNIF}[0, 1], b \in B)$.

     For $b \in B$, set $x_t^{(b)} = m_{t^{(b)}} x_0^{(b)} + \sqrt{v_{t^{(b)}}} z^{(b)}$ where $(z^{(b)} \sim \mathcal{N}(0, I), b \in B)$ are IID.

     Compute "score targets" $u_{t^{(b)}} = -z^{(b)}/\sqrt{v_{t^{(b)}}} = \nabla \log p_{t|0}(x_t^{(b)} \mid x_0^{(b)})$.

     Compute batch loss function

$$\mathcal{L}_B(\theta) = \frac{1}{|B|} \sum_{b \in B} \|s_\theta(x_t^{(b)}, t^{(b)}) - u_t^{(b)}\|^2.$$

     Update $\theta \leftarrow \theta - \eta \nabla_\theta \mathcal{L}(\theta)$.

**end while**

**Outputs:** $\widehat{\mu}, \widehat{\sigma}$ and trained score function $s_\theta^*$

---

---

**Algorithm 2** Generation of sample paths from trained model

---

**for** $j = 1, \ldots, N$ **do**

     Simulate discretized paths for the (reverse) SDE on the grid $(u_i = i/m, i = 0, \ldots, m)$.

     $Y_0 \sim N(m_T X_0, v_T I)$

     **for** $i = 0, \ldots, m$ **do**

$$Y_{u_{i+1}} = Y_{u_i} + \frac{1}{m}\left(\frac{1}{2}\beta_{T-u_i} Y_{u_i} + \beta_{T-u_i} s_\theta^*(Y_{u_i}, T - u_i)\right) + \sqrt{\beta_{T-u_i}/m}\, Z_i, \qquad Z_i \stackrel{iid}{\sim} N(0, I)$$

     **end for**

     $\widehat{\epsilon}_j \leftarrow Y_T$

**end for**

**for** $j = 1, \ldots N - 1$ **do** $\widehat{X_{t_{j+1}}} = \widehat{X_{t_j}} + \widehat{\mu}(\widehat{X_{t_j}})(t_{j+1} - t_j) + \widehat{\sigma}(\widehat{X_{t_j}})\widehat{\epsilon}_j$

**end for**

**return** Synthetic samples $\{\widehat{X_{t_k}}, k = 1, \ldots, N\}$

---

# 3 CONVERGENCE ANALYSIS

## 3.1 TV AND WASSERSTEIN CONVERGENCE

When examining the convergence of the reverse process, we start by making the following assumption regarding score matching:

**Assumption 1.** *For some $0 \leq t \leq T$, $\epsilon_{score} > 0$, we have access to score estimates $s_\theta(\cdot)$ satisfying $\mathbb{E}_{p_t}[\|s_\theta(X_t, t) - \nabla \log p_t(X_t, t)\|^2] \leq \epsilon_{score}^2$.*

De Bortoli et al. (2021) provided a first bound for $TV(\text{Law}(Y_T), p_0(\cdot))$, with the work of Chen et al. (2023) improving the bound to be polynomial in dimension $d$ and time $T$. From Assumption 1 and Chen et al. (2023), if we apply the total variation distance to our setting, we obtain

$$TV(\text{Law}(Y_T), p_0(\cdot)) \leq m_T \frac{\sqrt{\mathbb{E}_{p_0}[|X_0|^2]}}{2} + \epsilon_{\text{score}} \sqrt{\frac{T}{2}}. \tag{11}$$

We expand our convergence results by including Wasserstein bounds. First, we can make a stronger assumption on the score matching, i.e.

**Assumption 2.** *For some $0 \leq t \leq T$, $\epsilon_{score} > 0$, we have access to score estimates $s_\theta(\cdot)$ satisfying $\mathbb{E}_{p_t}[\|s_\theta(X_t, t) - \nabla \log p_t(X_t, t)\|_\infty] \leq \epsilon_{score}$.*

We require an additional assumption on the growth of the drift coefficient and regularity of the score function:

**Assumption 3.** *Consider the forward SDE equation 32. Then*

- *$\exists \ \rho(t) : [0, T] \to \mathbb{R}$ such that $(x - y)(f(x, t) - f(y, t)) \geq \rho(t)|x - y|^2$.*

- *Lipschitz score, i.e. $\exists \ L > 0$ such that $|\nabla \log p_t(x) - \nabla \log p_t(y)| \leq L|x - y|$.*

**Theorem 1** (Wasserstein bound on $\mathcal{W}_2^2(p_0, \text{Law}(Y_T))$). *Provided Assumptions 2 and 3 hold, and for hyperparameter $\lambda > 0$,*

$$\mathcal{W}_2^2(p_0, Law(Y_T)) \leq (e^{-\alpha_T} \mathbb{E}[\|x_0\|^2] + d(1 - \sqrt{1 - \exp(-\alpha_T)})^2)(e^{(1+2(L+\lambda))\alpha_T})$$
$$+ \frac{\epsilon_{score}^2}{2\lambda} \int_0^T \beta_t e^{(1+2(L+\lambda))\alpha_t} dt. \tag{12}$$

A derivation of the TV bound and proof of Theorem 1 are provided in Appendix B.

## 3.2 DECOMPOSING KERNEL AND DIFFUSION ERRORS

The reverse-time bounds above are stated for an idealized setting in which we have direct access to the "true" residual distribution. In practice, however, we do not observe the true drift and diffusion coefficients $\mu(x)$ and $a(x)$. Instead, we form nonparametric estimators $\widehat{\mu}(x)$ and $\widehat{a}(x)$ and construct residuals using the corresponding Cholesky factor $\widehat{\sigma}(x)$ of $\widehat{a}(x)$. To make this explicit, fix a time grid $t_k = k\Delta t$ and let the true residuals be

$$\epsilon_{t_k}^{(n)} = \sigma(X_{t_k}^{(n)})^{\top, -1} [\Delta X_{t_k}^{(n)} - \mu(X_{t_k}^{(n)})], \tag{13}$$

and the empirical residuals used in training be

$$\widehat{\epsilon}_{t_k}^{(n)} = \widehat{\sigma}(X_{t_k}^{(n)})^{\top, -1} [\Delta X_{t_k}^{(n)} - \widehat{\mu}(X_{t_k}^{(n)})]. \tag{14}$$

Let $p_0^{\text{res}}$ denote the law of the true residuals (restricted to the finite collection of increments used in the diffusion stage), and let $\widehat{p}_0^{\text{res}}$ denote the empirical law of the kernel-based residuals $\widehat{\epsilon}$. In the reverse-time analysis above, the initial law $p_0$ enters only through its second moment and its role as the starting distribution at time zero. In particular, Theorem 1 applies to *any* choice of initial law. We can therefore view the actual training procedure as applying Theorem 1 with $p_0 = \widehat{p}_0^{\text{res}}$, and then relate $p_0^{\text{res}}$ and $\widehat{p}_0^{\text{res}}$ via the triangle inequality in $\mathcal{W}_2$.

**Assumption 4** (Kernel residual approximation). *There exists a constant $\epsilon_{\text{ker}} \geq 0$ such that*

$$\mathcal{W}_2(p_0^{\text{res}}, \widehat{p}_0^{\text{res}}) \leq \epsilon_{\text{ker}}. \tag{15}$$

*Moreover, $\epsilon_{\text{ker}} \to 0$ as the number of observed paths and time steps tends to infinity under the high-frequency, large-sample regime used to motivate the kernel estimators $\widehat{\mu}$ and $\widehat{a}$.*

Assumption 4 is a compact way of summarizing the stage-one nonparametric error: it captures in a single quantity the combined effect of estimating the conditional mean and covariance and then mapping increments to residuals via the estimated Cholesky factor.

Let $p_0^{\mathrm{res}}$ be the law of the true residuals and $\widehat{p}_0^{\mathrm{res}}$ the law of the extracted residuals used in training. Suppose Assumption 2 holds with respect to the forward marginals of $\widehat{p}_0^{\mathrm{res}}$ and that Assumption 3 holds for the variance-preserving OU forward SDE. Let $\mathrm{Law}(Y_T^{\mathrm{ker}})$ denote the terminal law of the reverse-time SDE driven by the learned score network trained on $\widehat{p}_0^{\mathrm{res}}$. Then, under Assumption 4, we have

$$\mathcal{W}_2\big(p_0^{\mathrm{res}}, \mathrm{Law}(Y_T^{\mathrm{ker}})\big) \leq \epsilon_{\mathrm{ker}} \leq \mathcal{W}_2\big(\widehat{p}_0^{\mathrm{res}}, \mathrm{Law}(Y_T^{\mathrm{ker}})\big), \tag{16}$$

and consequently

$$\mathcal{W}_2^2\big(p_0^{\mathrm{res}}, \mathrm{Law}(Y_T^{\mathrm{ker}})\big) \leq 2\,\epsilon_{\mathrm{ker}}^2 \; + \; 2\,\mathcal{W}_2^2\big(\widehat{p}_0^{\mathrm{res}}, \mathrm{Law}(Y_T^{\mathrm{ker}})\big). \tag{17}$$

Furthermore, the second term on the right-hand side can be bounded by Theorem 1 with $p_0 = \widehat{p}_0^{\mathrm{res}}$, yielding

$$\mathcal{W}_2^2\big(p_0^{\mathrm{res}}, \mathrm{Law}(Y_T^{\mathrm{ker}})\big) \leq 2\epsilon_{\mathrm{ker}}^2 + 2\Big[\big(e^{-\alpha_T}\,\mathbb{E}_{\widehat{p}_0^{\mathrm{res}}}[\|x_0\|^2] + d\big(1 - \sqrt{1 - e^{-\alpha_T}}\big)^2\big)\,e^{(1+2(L+\lambda))\alpha_T}$$
$$\tag{18}$$

$$+ \frac{\epsilon_{\mathrm{score}}^2}{2\lambda} \int_0^T \beta_t e^{(1+2(L+\lambda))\alpha_t}\,dt\Big]. \tag{19}$$

*Proof.* The first inequality is the triangle inequality for $\mathcal{W}_2$:

$$\mathcal{W}_2\big(p_0^{\mathrm{res}}, \mathrm{Law}(Y_T^{\mathrm{ker}})\big) \leq \mathcal{W}_2\big(p_0^{\mathrm{res}}, \widehat{p}_0^{\mathrm{res}}\big) \; + \; \mathcal{W}_2\big(\widehat{p}_0^{\mathrm{res}}, \mathrm{Law}(Y_T^{\mathrm{ker}})\big), \tag{20}$$

and the second follows from the elementary inequality $(a + b)^2 \leq 2a^2 + 2b^2$ for $a, b \geq 0$. Assumption 4 identifies $\epsilon_{\mathrm{ker}}$ with the first term, and the bound in Theorem 1 applies exactly to the pair $\big(\widehat{p}_0^{\mathrm{res}}, \mathrm{Law}(Y_T^{\mathrm{ker}})\big)$, because the proof of Theorem 1 only requires Assumptions 2 and 3 to hold for the forward marginals of the initial law used in training. □

**Remark 3.** *An analogous decomposition holds in total variation. Let $\widehat{p}_T^{\mathrm{res}}$ denote the forward-time marginal obtained by evolving $\widehat{p}_0^{\mathrm{res}}$ under the OU forward SDE. Then*

$$TV\big(p_0^{\mathrm{res}}, \mathrm{Law}(Y_T^{\mathrm{ker}})\big) \leq TV\big(p_0^{\mathrm{res}}, \widehat{p}_0^{\mathrm{res}}\big) \; + \; TV\big(\widehat{p}_0^{\mathrm{res}}, \mathrm{Law}(Y_T^{\mathrm{ker}})\big), \tag{21}$$

*and the second term can be controlled by the same TV bound as in 11, with $p_0$ replaced by $\widehat{p}_0^{\mathrm{res}}$. In this way, stage-one kernel smoothing error appears explicitly as an additive term in both the $\mathcal{W}_2$ and total variation guarantees, rather than being implicitly absorbed into the score matching error.*

## 4 EXPERIMENTS

### 4.1 DETAILS OF NUMERICAL EXPERIMENTS

In the numerical experiment in this section, we will use the time-dependent "variance preserving" OU process from Section 1. We now assume that we have $N$ samples $\{x^n\}_{n=1}^N$ from our target distribution $p_0$. The empirical measure

$$\hat{p}_0 = \frac{1}{N} \sum_{n=1}^N \delta_{x^n} \tag{22}$$

is an approximation to $p_0$. If we start the forward SDE in $p_0$, we get marginals $\hat{p}_t$ defined below, where we apply the transition kernel to each data point in the empirical distribution $x^n$ at time 0 to $x_t$ and then average over all transition probabilities, as the empirical distribution at time $t$ can be approximated by the mean of the distributions resulting from diffusing each of the original $N$ data points according to the process:

$$\hat{p}_t(x_t) = \frac{1}{N} \sum_{n=1}^N p_{t|0}(x_t \mid x^n), \tag{23}$$

which is just a Gaussian mixture with $N$ components, one for each sample $x^n$. The components are centered at $m_t x^n$ and have variance $v_t$. These empirical marginals can actually be evaluated (unlike the unknown $p_t$). The reverse SDE is given by 8. We implement it using the Euler-Maruyama scheme. To advance the SDE by $\Delta t$, we compute the following iteration:

$$Y_{t_{i+1}} = Y_{t_i} + (t_{i+1} - t_i) \left( \frac{1}{2} \beta_{T-t} Y_{t_i} + \beta_{T-t} \nabla \log p_{T-t}(Y_{t_i}) \right) + \sqrt{\beta_{T-t}} Z_{t_{i+1}-t_i}, \qquad (24)$$

where $Z_{t_{i+1}-t_i}$ are independent with distribution $Z_{t_{i+1}-t_i} \sim \mathcal{N}(0, Z_{t_{i+1}-t_i} I)$. We will run the forward SDE until time $T = 1$. Then the time interval for the backward SDE is also $[0, T]$. We discretize this time interval into $(t_i)_{i=1}^{L}$, $t_0 = 0, t_L = 1$ and run the above scheme. We use $L = 1000$ steps of the reverse SDE; in practical applications, we might try to reduce the number of steps. Additionally, we use a geometric noise schedule for $\beta_t$:

$$\beta_t = \beta_{\max}^{1-t} \beta_{\min}^{t} = \beta_{\max} \left( \frac{\beta_{\min}}{\beta_{\max}} \right)^{t}. \qquad (25)$$

In practice, we discretize over $R = 10$ steps, so that

$$\beta_r = \beta_{\max} \left( \frac{\beta_{\min}}{\beta_{\max}} \right)^{\frac{r}{R-1}}, \qquad (26)$$

for $r = 0, \ldots, R - 1$. We can now plug in the empirical drift $\nabla \log \hat{p}_t$ into the reverse SDE and run it. The result is the exact reverse SDE for the data distribution $p_0 = \hat{p}_0$. Recall that we can exactly recover $\hat{p}_0$. Since $p_{t,0}$ is Gaussian we can evaluate the gradient as

$$\nabla \log p_{t,0}(x \mid x_0) = \nabla \log \left( (2\pi v_t)^{-d/2} \exp \left( -\frac{\|x - m_t x_0\|^2}{2v_t} \right) \right) \qquad (27)$$

$$= \nabla \left[ -\frac{d}{2} \log (2\pi v_t) - \frac{\|x - m_t x_0\|^2}{2v_t} \right] \qquad (28)$$

$$= -\frac{(x - m_t x_0)}{v_t}. \qquad (29)$$

Since we do not have access to $\nabla \log \hat{p}_t$, we approximate it using a neural network and 47. The objective is 50, and if we let

$$\bar{L}(\theta, t) = \mathbb{E}_{x_0 \sim \hat{p}_{\text{data}}} \mathbb{E}_{x \sim p_{t|0}(x|x_0)} \left[ \|\nabla \log p_{t|0}(x \mid x_0) - s_\theta(x, t)\|^2 \right], \qquad (30)$$

then we need to optimize the network for all $t$, not just one specific $t$, and therefore use

$$\bar{L}(\theta) = \mathbb{E}_{t \sim U[0,1]}[\bar{L}(\theta, t)]. \qquad (31)$$

This loss can now be approximated by randomly choosing data points from the training batch (as samples from $\hat{p}_0$ and also randomly generating times $t \sim \mathcal{U}[0, 1]$).

The score-based diffusion model is a four layer feed-forward network, and it consists of a linear projection with a GELU activation and a learnable embedding layer, followed by a three layer feed-forward network with dropout-regularized GELU activations. Optimization is Adam (learning rate $5 \times 10^{-3}$), batch size is 128, and training is run for 10000 iterations. Reverse-time sampling uses Euler-Maruyama with step sizes scaled as $u_i$ and $T_{\text{emp}} = 200$ inner steps. It is trained on the filtered residuals using denoising score matching and the *exact* Gaussian conditional target for the marginals.

## 4.2 SYNTHETIC MULTIVARIATE TIME SERIES

For our first experiment, we test a multivariate time series – a vector AR(1) process where a mixture of Gaussians generates the innovations. We define $\phi = \{\phi_1, \phi_2\} \in \mathbb{R}^{d \times d}$ to be the AR coefficient matrix. Then we define $\varepsilon_t \sim \sum_{k=1}^{K} \pi_k \mathcal{N}(\mu_k, \Sigma_k)$ to be the innovations, where $\mu_k \in \mathbb{R}^d$ and $\Sigma_k \in \mathbb{R}^{d \times d}$ are the mean and covariance for each mixture component $k = 1, \ldots, K$. Therefore,

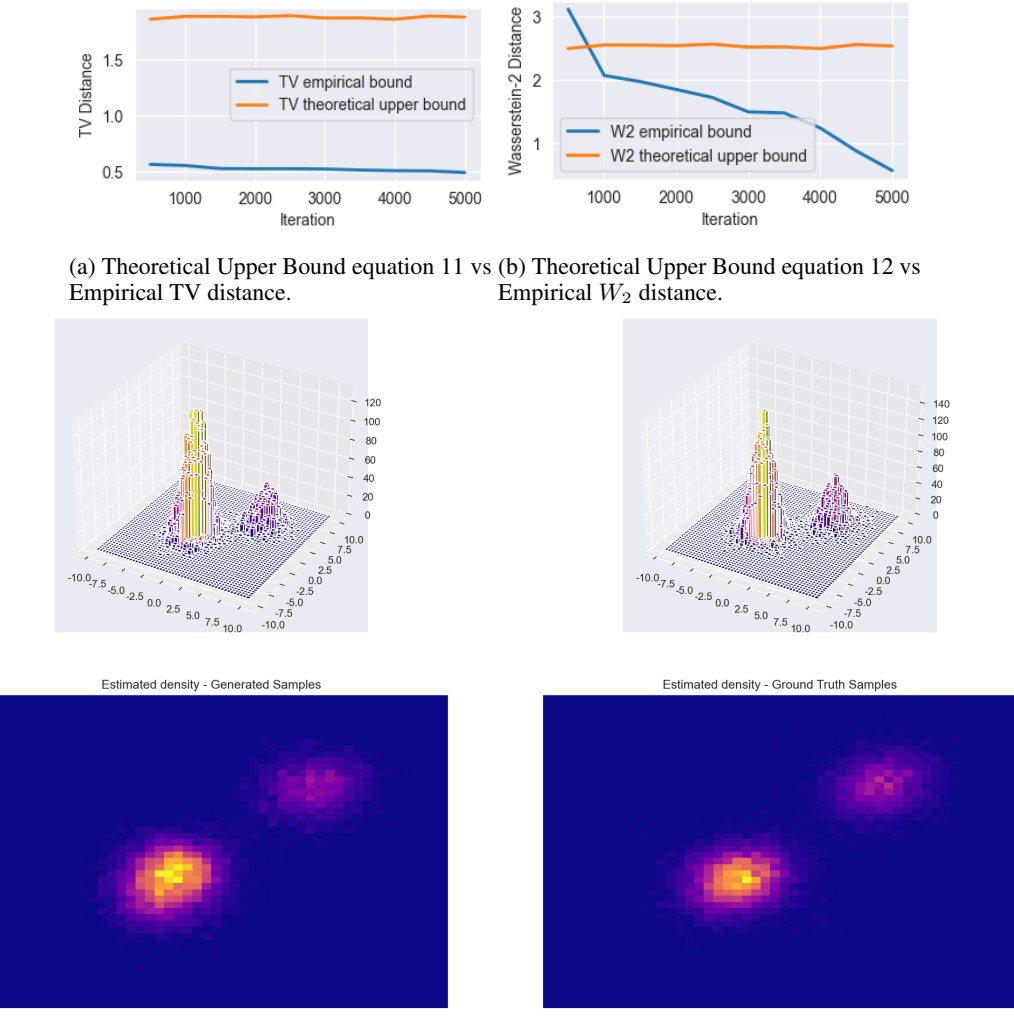

(a) Theoretical Upper Bound equation 11 vs Empirical TV distance.

(b) Theoretical Upper Bound equation 12 vs Empirical $W_2$ distance.

(c) Density plots of ground truth residuals versus residuals generated by the score-based diffusion model.

Figure 1: 1a and 1b are plots for the theoretical versus empirical Total Variation distance and Wasserstein-2 distance through training iterations. 1c shows 3d surface plots and heat maps of the generated residuals (left) versus the ground truth residuals (right).

each path evolves as $X_t = \phi X_{t-1} + \varepsilon_t$. We simulate data in $d = 20, 30, 50$ dimensions with $T = 1000, 2000, 5000$ time steps.

In Figure 1, we report (i) empirical Total Variation and Wasserstein-2 bounds between ground truth and generated residuals, and (ii) plots of the first two components of the ground truth and generated residuals, while Figure 2 shows scaling plots for $d = 20, 30, 50$ dimensions and $T = 1000, 2000, 5000$ time steps. We do note some metric-dependent behavior; $W_2$ is dominated by matching low-order moments and overall mass transportation cost. As the score network learns, these improve steadily, hence the clear decreasing trend. TV is more sensitive to localized density mismatches and tail behavior, which are harder to estimate reliably in high dimension from finite samples; its empirical estimator thus has higher variance. In our implementation, the TV estimator is based on a plug-in approach using a finite number of samples and bins; for large $d$ this can be noisy. The theoretical upper bound 11 is driven by score error and noise schedule and is not tight in finite-sample TV.

Table 1 shows results of lag and bandwidth sensitivity studies, and Table 2 probes Cholesky factor ambiguity. In particular, we conducted:

- We conducted a lag-sensitivity study, conditioning the kernel estimator on $[X_t, X_{t-1}]$, and $[X_t, X_{t-1}, X_{t-2}]$. The $L_2$ norm of the conditional mean decreased from 0.853 to 0.725 and the mean residual standard deviation from 2.997 to 2.961, indicating only mild gains beyond a first-order Markov state. Thus, the original Markov assumption is empirically sound in our setup.

- We performed a bandwidth sensitivity study for the multimodal kernel estimator on the AR-mixture process. For bandwidths 0.25, 0.5, 1.0, the $L_2$ change in $\hat{\mu}$ relative to the reference (0.25) is 0.52 and 0.72, indicating moderate smoothing effects, but the residual skewness and kurtosis remain stable. This suggests that the residual distribution's non-Gaussian features are robust to bandwidth choice.

- We probed the Cholesky ambiguity by rotating the residuals with random orthogonal matrices and comparing covariances. For several rotations ($r = 0, 3$), the covariance stays very close to the original $\|\Sigma_r - \Sigma_{\text{base}}\|_F \ll \|\Sigma_{\text{base}}\|_F$, where $\|\widehat{\Sigma}_r\|_F$ is the Frobenius norm of the covariance implied by rotation $r$ and $\|\widehat{\Sigma}_{\text{base}}\|_F$ is the Frobenius norm of the base covariance (constant across $r$). However, extreme rotations ($r = 2, 4$) can change it more substantially. Since our implementation uses a fixed Cholesky convention, the diffusion model always sees a single, consistent residual distribution, and our experiments indicate that its second-order geometry is reasonably stable under typical rotations.

Table 1: Kernel lag and bandwidth sensitivity diagnostics

(a) Lag sensitivity (fixed bandwidth)

| Lag $L$ | $\|\hat{\mu}\|_{L^2}$ | mean std($\epsilon$) |
|---|---|---|
| 1 | 0.853 | 2.997 |
| 2 | 0.767 | 2.965 |
| 3 | 0.725 | 2.961 |

(b) Bandwidth sensitivity

| $h$ | $\|\hat{\mu} - \mu_{\text{ref}}\|_{L^2}$ | $\mathbb{E}[\text{skew}(\epsilon)]$ | $\max \text{skew}(\epsilon)$ | $\mathbb{E}[\text{kurt}(\epsilon)]$ | $\max \text{kurt}(\epsilon)$ |
|---|---|---|---|---|---|
| 0.25 | 0.000 | 0.625 | 0.635 | -0.760 | 0.790 |
| 0.50 | 0.518 | 0.685 | 0.688 | -0.863 | 0.879 |
| 1.00 | 0.723 | 0.699 | 0.699 | -0.910 | 0.920 |

Table 2: Cholesky factor ambiguity: effect on implied covariance

| Rotation index $r$ | $\|\widehat{\Sigma}_r\|_F$ | $\|\widehat{\Sigma}_r - \widehat{\Sigma}_{\text{base}}\|_F$ | $\|\widehat{\Sigma}_{\text{base}}\|_F$ |
|---|---|---|---|
| 0 | 11.967 | 0.744 | 12.711 |
| 1 | 10.453 | 2.258 | 12.711 |
| 2 | 33.594 | 20.883 | 12.711 |
| 3 | 12.905 | 0.194 | 12.711 |
| 4 | 2.138 | 10.572 | 12.711 |

Table 3 shows expectations of test functionals $f(\widehat{X})$ as targeted probes of the generated samples against analytic oracles computed directly from the (known) data generating process along with their Monte Carlo standard errors. In particular, we utilize 3 test functionals:

- `max_component`: to test extreme value behavior across dimensions,

- `basket`: a linear average across dimensions to test first moment, and

- `basket_put`: $\max(K - \text{basket}, 0)$ to probe tail behavior.

To assess sampling variability, we subsample $n$ draws from the model repeatedly and check that both the empirical standard deviation and the within-batch standard errors scale like $\frac{1}{\sqrt{n}}$. Thus, we include standard deviation $\times \sqrt{n}$ in the table and show it is roughly constant across $n$.

Table 3: Oracle versus expectations of test functionals

| Functional | $n$ | Oracle | Model Mean | MC Std. Error | Std. Deviation $\times \sqrt{n}$ |
|---|---|---|---|---|---|
| max_component | 2000 | 1.588 | 1.862 | 0.071 | 0.112 |
| | 4000 | 1.588 | 1.868 | 0.050 | 0.116 |
| | 8000 | 1.588 | 1.860 | 0.035 | 0.118 |
| basket | 2000 | -7.988 | -11.050 | 0.052 | 0.140 |
| | 4000 | -7.988 | -11.051 | 0.037 | 0.125 |
| | 8000 | -7.988 | -11.055 | 0.026 | 0.137 |
| basket_put | 2000 | 107.988 | 111.054 | 0.039 | 0.109 |
| | 4000 | 107.988 | 111.051 | 0.027 | 0.119 |
| | 8000 | 107.988 | 111.050 | 0.019 | 0.117 |

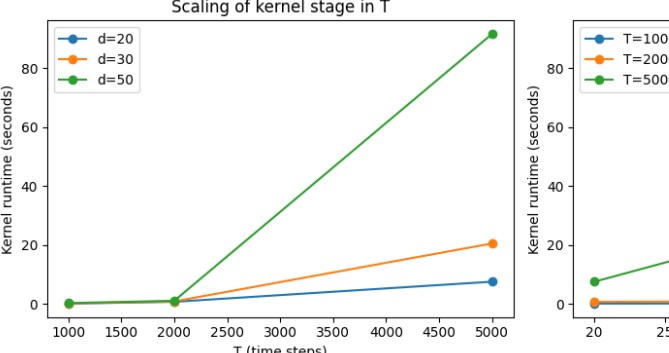 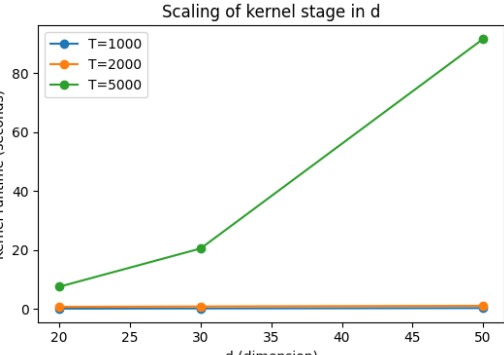

Figure 2: Scaling curves across $T = 1000, 2000, 5000$ time steps and $d = 20, 30, 50$ dimensions.

## 5 DISCUSSION AND FUTURE WORK

Our study focuses on generating high-dimensional processes, and the convergence results derived under strong regularity assumptions. Empirical TV and $W_2$ distances were upper-bounded by their theoretical bounds, with deviations decreasing over training iterations, suggesting our convergence estimates are informative in practice. The agreement of expectations of the test functionals with their analytic oracles demonstrates the method preserves essential first and second-order structure. The surface plots confirm that the generated residuals capture the geometry of the ground-truth residuals. Notably, the model successfully recovers multimodal residual distributions.

Further work is required to assess robustness as well as comparison to baselines such as time-series DDPMs, latent-SDE, and conditional diffusion. Additionally, conducting stress tests where the kernel stage is misspecified, rare-event checks, and specifying downstream tasks would help expand benchmarks/evaluation.

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

## A    BACKGROUND ON SCORE-BASED DIFFUSION MODELS

In Section 1, we introduced the idea of time-reversed diffusions. Below, we state the property for clarity. Consider the following well-defined SDE:

$$
\begin{aligned}
dX_t &= f(X_t, t)dt + g(X_t, t)dW_t \\
X_0 &\sim p_0
\end{aligned}
\tag{32}
$$

$f$ and $g$ satisfy local Lipschitz continuity and linear growth conditions, so the existence of $p_t$ is guaranteed. Additionally, $p_t$ is differentiable and strictly positive, provided that $g(x,t)g(x,t)^\top$ is positive definite. Starting from the density $p_T$, we expect that running $X$ in reverse time would generate samples from the density $p_0$. This time reversal property of diffusions is a well-known fact in stochastic analysis (Anderson (1982), Haussmann & Pardoux (1986), Föllmer (2005)).

**Proposition 1** (Time Reversal Haussmann & Pardoux (1986))**.** *Consider the SDE 32. Let $Y_t = X_{T-t}$ for $t \in [0, T]$, $T > 0$. Then, under the conditions outlined above, $Y$ is a diffusion process with drift given by*

$$
\tilde{f}(x, t) = -f(x, T-t) + \frac{\mathrm{div}(p_{T-t}(x) \cdot a(x, T-t))}{p_{T-t}(x)},
\tag{33}
$$

*where $a(x, t) = g(x, t)g(x, t)^\top$. Expanding the divergence term component-wise,*

$$
(\mathrm{div}(p_{T-t}(x) \cdot a(x, T-t)))^i = \sum_{j=1}^{d} \frac{\partial}{\partial x^j}(p_{T-t}(x)a^{ij}(x, T-t))
\tag{34}
$$

$$
= \sum_{j=1}^{d}\left[\frac{\partial p_{T-t}(x)}{\partial x^j}a^{ij}(x, T-t) + p_{T-t}(x)\frac{\partial a^{ij}(x, T-t)}{\partial x^j}\right], \tag{35}
$$

*leads to the vector form*

$$\operatorname{div}(p_{T-t}(x) \cdot a(x, T-t)) = p_{T-t}(x) \operatorname{div} a(x, T-t) + a(x, T-t)\nabla p_{T-t}(x). \quad (36)$$

*Then*

$$\tilde{f}(x,t) = -f(x, T-t) + \operatorname{div} a(x, T-t) + a(x, T-t)\nabla \log(p_{T-t}(x)) \quad (37)$$

*satisfying*

$$\begin{aligned} \mathrm{d}Y_t &= \tilde{f}(Y_t, t)\mathrm{d}t + g(Y_t, T-t)\mathrm{d}\bar{W}_t \\ Y_0 &\sim p_T. \end{aligned} \quad (38)$$

*Running the backward procedure will generate $Y_T \sim p_0$ at time $T$.*

We note a few issues that arise if we want to run the reverse process: we do not have sample access to $p_T$ the initial condition of the reverse SDE, and we do not know $p_t$, which means we do not know the drift $\nabla \log p_{T-t}$. The easiest way to deal with the initial condition is to consider choosing $f$ and $g$ such that $X_t$ converges to a prior distribution $p_\infty$. This allows the initial distribution of the reverse process to be $Y_0 \sim p_\infty$. We want $p_T$ and $p_\infty$ to be sufficiently close, so that the distribution of $X_T$ is close to $p_0$. In practice, we choose the parameters so that the distribution $p_\infty$ is Gaussian. Then we only need to compute $\nabla \log p_{T-t}$.

The task of estimating the score function $\nabla \log p_t$ (Ho et al. (2020), Song & Ermon (2019), Song et al. (2021)) is **score matching**, and it involves reducing the estimation of the score function to a supervised learning task. Score matching dates back to Tweedie's Formula from the '50s Efron (2011). Essentially, we will see that estimating $\nabla \log p_t$ is equivalent to estimating the noise added.

**Proposition 2** (Tweedie's Formula). *Given $\tilde{x} = x + e$ for $x \sim p$ and $e \sim \mathcal{N}(0, \sigma^2 \cdot I)$,*

$$\mathbb{E}[x \mid \tilde{x}] = \tilde{x} + \sigma^2 \cdot \nabla \log \tilde{p}(\tilde{x})$$

*where $\tilde{p}$ is the density for $\tilde{x}$.*

*Proof.* Since $e \sim \mathcal{N}(0, \sigma^2 I)$, the density of $\tilde{x}$ is:

$$\tilde{p}(\tilde{x}) = \int p(x) \cdot \rho_\sigma(\tilde{x} - x)\, dx, \quad (39)$$

where $\rho_\sigma(z) \propto \exp\left(-\frac{z^2}{2\sigma^2}\right)$ is a Gaussian with variance $\sigma^2$. The posterior expectation of $x$ given $\tilde{x}$ is:

$$\mathbb{E}[x \mid \tilde{x}] = \frac{\int x\, p(x)\, \rho_\sigma(\tilde{x} - x)\, dx}{\int p(x)\, \rho_\sigma(\tilde{x} - x)\, dx}. \quad (40)$$

Taking the gradient of $\rho_\sigma(\tilde{x} - x)$ with respect to $\tilde{x}$:

$$\nabla_{\tilde{x}}\rho_\sigma(\tilde{x} - x) = \frac{x - \tilde{x}}{\sigma^2}\rho_\sigma(\tilde{x} - x). \quad (41)$$

Differentiating the log of $\tilde{p}(\tilde{x})$:

$$\nabla_{\tilde{x}} \log \tilde{p}(\tilde{x}) = \frac{\int \frac{x-\tilde{x}}{\sigma^2}p(x)\rho_\sigma(\tilde{x} - x)\, dx}{\int p(x)\rho_\sigma(\tilde{x} - x)\, dx}, \quad (42)$$

which simplifies to:

$$\nabla_{\tilde{x}} \log \tilde{p}(\tilde{x}) = \frac{\mathbb{E}[x \mid \tilde{x}] - \tilde{x}}{\sigma^2}. \quad (43)$$

Rearranging this equation yields Tweedie's formula:

$$\mathbb{E}[x \mid \tilde{x}] = \tilde{x} + \sigma^2 \nabla \log \tilde{p}(\tilde{x}). \quad (44)$$

$$\square$$

We can consider $\nabla \log \tilde{p}(\tilde{x})$ as the Bayes optimal estimate of the noise – hence given a noisy sample $X_t$, the supervised learning task is to predict the noise added. In the following definitions, we formalize the concept of score matching. We assume a collection of score estimates $\{s_\theta(x, t)\}$ on $\mathbb{R}^d \times \mathbb{R}_+$ parameterized by $\theta$ – typically a neural network. The objective is to solve the following optimization problem:

$$\min_\theta \mathbb{E}_{p_t}[\|\nabla \log p_t(X_t, t) - s_\theta(X_t, t)\|^2]. \quad (45)$$

This is not possible to calculate as we do not know $\nabla \log p_t(X_t, t)$. An alternative approach is that of **implicit score matching**.

**Definition 1** (Implicit Score Matching). *Hyvärinen (2005) We compute*

$$\min_{\theta} \mathbb{E}_{p_t}[\|s_\theta(X_t, t)\|^2 + 2\nabla s_\theta(X_t, t)]. \tag{46}$$

However, implicit score matching may be computationally complex if the dimension $d$ is very large – gradient descent methods would not be efficient as the computation of the gradient of the score network scales linearly in the dimension. The method of **denoising score matching** is one possible approach when working with high-dimensional data.

**Definition 2** (Denoising Score Matching). *Vincent (2011) We condition $X_t$ on $X_0$, replacing $\nabla \log p_t(X_t, t)$ with $\nabla \log p_{t|0}(X_t \mid X_0)$:*

$$\min_{\theta} \mathbb{E}_{x_0 \sim p_{data}} \mathbb{E}_{x \sim p_{t|0}(x|x_0)}[\|\nabla \log p_{t|0}(x \mid x_0) - s_\theta(x, t)\|^2]. \tag{47}$$

To show the equivalence between 45 and 47, we start with the standard objective, expanding the squared norm:

$$\mathbb{E}_{p_t}\left[\|\nabla \log p_t(X_t) - s_\theta(X_t, t)\|^2\right] = \mathbb{E}_{p_t}\left[\|\nabla \log p_t(X_t)\|^2\right] - 2\,\mathbb{E}_{p_t}\left[\langle \nabla \log p_t(X_t), s_\theta(X_t, t)\rangle\right]$$
$$+ \mathbb{E}_{p_t}\left[\|s_\theta(X_t, t)\|^2\right]. \tag{48}$$

Now, we note that the marginal score in the cross-term can be replaced by the conditional score:

$$\mathbb{E}_{p_t}\left[\langle \nabla \log p_t(X_t), s_\theta(X_t, t)\rangle\right] = \mathbb{E}_{x_0 \sim p_0}\mathbb{E}_{x \sim p_{t|0}(x|x_0)}\left[\langle \nabla \log p_{t|0}(x \mid x_0), s_\theta(x, t)\rangle\right]. \tag{49}$$

Given that $\mathbb{E}_{p_t}\left[\|\nabla \log p_t(X_t)\|^2\right]$ and $\mathbb{E}_{p_t}\left[\|s_\theta(X_t, t)\|^2\right]$ are both unaffected by the conditioning on $X_0$ directly, we can rewrite the entire objective incorporating this conditioning:

$$\mathbb{E}_{p_t}\left[\|\nabla \log p_t(X_t) - s_\theta(X_t, t)\|^2\right] = \mathbb{E}_{x_0 \sim p_0}\mathbb{E}_{x \sim p_{t|0}(x|x_0)}\left[\|\nabla \log p_{t|0}(x \mid x_0) - s_\theta(x, t)\|^2\right], \tag{50}$$

which is exactly the denoising score matching objective. To reiterate, the goal of denoising score matching is to show that the score function of some "noisy" sample should move to a clean sample gradually. We saw that the conditional distribution $p_{t|0}(X_t \mid X_0)$ should be something simple, ideally Gaussian.

# B    PROOFS OF CONVERGENCE

From Assumption 1 and Chen et al. (2023), we have

$$TV(\text{Law}(Y_T), p_0(\cdot)) \leq TV(p(T, \cdot), p_{\text{noise}}(\cdot)) + \epsilon_{\text{score}}\sqrt{\frac{T}{2}}. \tag{51}$$

Recall the time-$t$ transition kernel is given by

$$p_{t|0}(\cdot \mid X_0 = x_0) = \mathcal{N}(m_t x_0, v_t I). \tag{52}$$

In order to quantify $TV(p(T, \cdot), p_{\text{noise}}(\cdot))$, we use KL divergence $KL(\mathcal{N}(m_t X_0, v_t I) \| \mathcal{N}(0, I))$ and Pinsker's inequality:

$$\frac{1}{2}\left(\text{Tr}(I^{-1}v_T I) + (0 - m_T X_0)^\top I^{-1}(0 - m_T X_0) - d + \log\left(\frac{\det I}{\det(v_T I)}\right)\right) \tag{53}$$

$$= \frac{1}{2}(v_T d + m_T^2 |X_0|^2 - d - d\log(v_T))) \tag{54}$$

$$= \frac{1}{2}(m_T^2 |X_0|^2 - d(1 - v_T + \log(v_T))) \tag{55}$$

$$= \frac{1}{2}(m_T^2 |X_0|^2 - d(m_T^2 + \log(v_T))) \tag{56}$$

$$\leq \frac{1}{2}m_T^2 X_0^2 \text{ as } T \to \infty. \tag{57}$$

Thus, $\mathbb{E}_{p_0}[KL(\mathcal{N}(m_t X_0, v_t I) \| \mathcal{N}(0, I))] \leq \frac{1}{2} m_T^2 X_0^2$, so that

$$TV(p(T, \cdot), p_{\text{noise}}(\cdot)) \leq \sqrt{\frac{1}{4} m_T^2 \mathbb{E}_{p_0}[|X_0|^2]} \leq m_T \frac{\sqrt{\mathbb{E}_{p_0}[|X_0|^2]}}{2}. \tag{58}$$

Therefore, the complete inequality is

$$TV(\text{Law}(Y_T), p_0(\cdot)) \leq m_T \frac{\sqrt{\mathbb{E}_{p_0}[|X_0|^2]}}{2} + \epsilon_{\text{score}} \sqrt{\frac{T}{2}}. \tag{59}$$

**Remark 4.** *The term $m_T$ in the above bound depends on the integrated noise schedule $\alpha_T = \int_0^T \beta_s \mathrm{d}s$ via $m_T = \exp(-\frac{1}{2}\alpha_T)$. For the variance-preserving OU schedule used in our experiments, where $\beta_t$ is positive and bounded away from zero on $[0, T]$, $\alpha_T$ grows at least linearly in $T$ and hence $m_T$ decays at least exponentially in $T$. The first term on the right-hand side of the TV bound 51 therefore behaves like $\exp(-\frac{1}{2}\alpha_T) \cdot \frac{\sqrt{\mathbb{E}_{p_0}[|X_0|^2]}}{2}$. The second term grows only like $\epsilon_{score} \sqrt{\frac{T}{2}}$. This makes precise the trade–off between the choice of noise schedule, which controls how quickly the forward process forgets its initialization, and the accuracy with which the learned score approximates the true score along the reverse path.*

**One-sided Lipschitz condition.** We see the one-sided Lipschitz condition in Assumption 3 holds for our particular OU SDE, i.e. when $\rho(t) = -\frac{\beta_t}{2}$:

$$f(x, t) - f(y, t) = -\frac{1}{2} \beta_t (x - y) \tag{60}$$

$$(x - y) \cdot (f(x, t) - f(y, t)) = -\frac{1}{2} \beta_t (x - y)^2 = -\frac{1}{2} \beta_t \|x - y\|^2, \tag{61}$$

so the inequality

$$(x - y)(f(x, t) - f(y, t)) \geq \rho(t)|x - y|^2 \tag{62}$$

holds with equality when $\rho(t) = -\frac{\beta_t}{2}$. Since $\beta_t \geq \beta_{\min} > 0$, we have $\rho(t) \leq -\frac{\beta_t}{2} < 0$, i.e. the drift is contractive in the one-sided Lipschitz sense.

**Lipschitz score assumption.** We also show Lipschitz score for the synthetic data setting outlined in Section 4. To begin, we assume the residuals at time 0 have a finite Gaussian mixture law

$$p_0(x) = \sum_{k=1}^{K} \pi_k \mathcal{N}(x; \mu_k, \Sigma_k), \tag{63}$$

where $\pi_k > 0$ and $\sum_{k=1}^{K} \pi_k = 1$, $\mu_k \in \mathbb{R}^d$, $\Sigma_k \in \mathbb{R}^{d \times d}$ are symmetric positive definite, and all eigenvalues of $\Sigma_k$ lie in $[\lambda_{\min}, \lambda_{\max}]$ for some fixed $0 < \lambda_{\min} \leq \lambda_{\max} < \infty$. Let $X_t$ satisfy 32 with $\beta_t$ continuous and bounded on $[0, T]$, and $\beta_t \geq \beta_{\min} > 0$. Define $\alpha_t = \int_0^t \beta_s \, ds$, $m_t = \exp(-\alpha_t/2)$, $v_t = 1 - \exp(-\alpha_t)$ as before. Then, conditional on $X_0 = x_0$, we have:

$$X_t \,\big|\, X_0 = x_0 \ \sim \ \mathcal{N}(m_t x_0, \ v_t I). \tag{64}$$

For each mixture component $k$, we can track how it evolves: if at time zero

$$X_0 \,\big|\, (\text{component } k) \sim \mathcal{N}(\mu_k, \Sigma_k),$$

then

$$X_t \,\big|\, (\text{component } k) \sim \mathcal{N}(m_t \mu_k, \ \Sigma_k(t)),$$

with

$$\Sigma_k(t) \ = \ m_t^2 \Sigma_k + v_t I. \tag{65}$$

Because $\Sigma_k$ has eigenvalues in $[\lambda_{\min}, \lambda_{\max}]$ and $v_t > 0$ for all $t > 0$, the eigenvalues of $\Sigma_k(t)$ stay in a compact interval $[\underline{\lambda}(t), \overline{\lambda}(t)]$ with $\underline{\lambda}(t) > 0$.

So for any fixed $t > 0$:

$$p_t(x) \ = \ \sum_{k=1}^{K} \pi_k \mathcal{N}(x; m_t \mu_k, \Sigma_k(t)) \tag{66}$$

is again a finite Gaussian mixture with non-degenerate (strictly positive definite) covariances. For each component $k$ at time $t$, the score is:

$$s_k(x,t) = \nabla_x \log \mathcal{N}(x; m_t \mu_k, \Sigma_k(t)) = -\Sigma_k(t)^{-1}(x - m_t \mu_k). \tag{67}$$

This is an affine function in $x$, with constant Jacobian:

$$\nabla_x s_k(x,t) = -\Sigma_k(t)^{-1}. \tag{68}$$

Now we define the mixture density:

$$p_t(x) = \sum_{k=1}^{K} \pi_k \phi_k(x,t), \tag{69}$$

with $\phi_k(x,t) = \mathcal{N}(x; m_t \mu_k, \Sigma_k(t))$. The mixture posterior weights are:

$$w_k(x,t) = \frac{\pi_k \phi_k(x,t)}{p_t(x)}. \tag{70}$$

Then the mixture score is:

$$s_t(x) = \nabla_x \log p_t(x) = \sum_{k=1}^{K} w_k(x,t)\, s_k(x,t). \tag{71}$$

We can check this by differentiating:

$$\nabla_x p_t(x) = \sum_{k=1}^{K} \pi_k \nabla_x \phi_k(x,t) = \sum_{k=1}^{K} \pi_k \phi_k(x,t)\, s_k(x,t), \tag{72}$$

so

$$s_t(x) = \frac{1}{p_t(x)} \nabla_x p_t(x) = \sum_{k=1}^{K} \frac{\pi_k \phi_k(x,t)}{p_t(x)} s_k(x,t) = \sum_{k=1}^{K} w_k(x,t)\, s_k(x,t). \tag{73}$$

To show the score $s_t(x)$ is globally Lipschitz in $x$, we want to show the Hessian of $\log p_t(x)$ is bounded:

$$\nabla_x^2 \log p_t(x) \text{ has bounded operator norm for all } x \quad \Rightarrow \quad s_t(x) \text{ is Lipschitz.}$$

A convenient formula is:

$$\nabla_x^2 \log p_t(x) = \frac{\nabla_x^2 p_t(x)}{p_t(x)} - \frac{\nabla_x p_t(x)\, \nabla_x p_t(x)^{\top}}{p_t(x)^2}. \tag{74}$$

We know:

$$\nabla_x p_t(x) = \sum_{k=1}^{K} \pi_k \phi_k(x,t)\, s_k(x,t) \tag{75}$$

$$\nabla_x^2 p_t(x) = \sum_{k=1}^{K} \pi_k \nabla_x^2 \phi_k(x,t). \tag{76}$$

For each Gaussian component, $\phi_k(x,t)$ is smooth and its derivatives decay like a polynomial in $\|x\|$ times $\exp(-c\|x\|^2)$. The second derivatives $\nabla_x^2 \phi_k(x,t)$ involve terms of the form

$$\phi_k(x,t) \left( A_k(t) + B_k(t)(x - m_t \mu_k)(x - m_t \mu_k)^{\top} \right) \tag{77}$$

for some bounded matrices $A_k(t), B_k(t)$ depending on $\Sigma_k(t)$. Because $\Sigma_k(t)$ is uniformly non-degenerate (eigenvalues bounded above and below for $t$ in a compact interval away from 0), those matrices are uniformly bounded in operator norm. Combining:

- $p_t(x)$ is a finite sum of Gaussian densities with non-degenerate covariances, so $p_t(x) > 0$ for all $x$, and it decays at least like $\exp(-c\|x\|^2)$ at infinity.

- $\nabla_x p_t(x)$ and $\nabla_x^2 p_t(x)$ are finite Gaussian mixtures of polynomials times Gaussians, so they are bounded by constants times $\exp(-c\|x\|^2)$ and $\exp(-c\|x\|^2)\|x\|^2$, respectively.

It follows that each term

$$\frac{\nabla_x^2 p_t(x)}{p_t(x)}, \quad \frac{\nabla_x p_t(x)\,\nabla_x p_t(x)^\top}{p_t(x)^2}$$

is bounded in operator norm uniformly in $x$, for each fixed $t > 0$. This is a standard property of Gaussian mixtures with strictly positive-definite covariances. So for each fixed $t > 0$, there exists a finite constant $L_t$ such that

$$\sup_{x \in \mathbb{R}^d} \left\| \nabla_x^2 \log p_t(x) \right\|_{\mathrm{op}} \leq L_t. \tag{78}$$

Hence the score is globally Lipschitz:

$$\|s_t(x) - s_t(y)\| \leq L_t \|x - y\| \quad \text{for all } x, y \in \mathbb{R}^d. \tag{79}$$

Now consider $t$ in a compact time interval $[t_0, T]$ with $t_0 > 0$. On this interval:

- $\beta_t$ is bounded above and below, so $\alpha_t$ and thus $m_t, v_t$ are continuous and bounded.

- $v_t = 1 - \exp(-\alpha_t)$ has a strictly positive lower bound $v_{\min} > 0$ for $t \geq t_0$.

Therefore the eigenvalues of each

$$\Sigma_k(t) = m_t^2 \Sigma_k + v_t I \tag{80}$$

are uniformly bounded between strictly positive constants for all $t \in [t_0, T]$, and so are the norms of $\Sigma_k(t)^{-1}$. This implies all the constants that appear in the derivative bounds above can be chosen independent of $t$ on that interval. So there exists a finite constant $L$ such that

$$\sup_{t \in [t_0, T]} \sup_{x \in \mathbb{R}^d} \left\| \nabla_x^2 \log p_t(x) \right\|_{\mathrm{op}} \leq L. \tag{81}$$

In particular, for all $t \in [t_0, T]$ and all $x, y$:

$$\|s_t(x) - s_t(y)\| \leq L \|x - y\|. \tag{82}$$

That is exactly the global Lipschitz score condition you assume in the $W_2$ convergence theorem.

Finally, to prove Theorem 1, we proceed by using coupled SDEs and a Grönwall-type argument. We will construct a coupling between $A_t$, the exact reverse-time diffusion (which uses the true score) and $B_t$, the approximate reverse-time diffusion (which uses the learned score). Then we can bound the Wasserstein-2 distance by

$$\mathcal{W}_2(p_0, \mathrm{Law}(Y_T))^2 \leq \mathbb{E}[\|A_T - B_T\|^2]. \tag{83}$$

We consider the same Brownian motion $W_t$ and define $A_0 \sim p_T$, $B_0 \sim p_{\mathrm{noise}}$. We have the following coupled SDEs:

$$\begin{cases} dA_t = [-f(A_t, T-t) + g^2(T-t)\nabla \log p_{T-t}(A_t)]dt + g(T-t)dW_t \\ dB_t = [-f(B_t, T-t) + g^2(T-t)s_\theta(B_t, T-t)]dt + g(T-t)dW_t \end{cases} \tag{84}$$

Define the coupling error by

$$\delta_t := \mathbb{E}[\|A_t - B_t\|^2]. \tag{85}$$

Applying Itó's formula yields

$$\frac{d}{dt}\delta_t = 2\mathbb{E}[(A_t - B_t)(\tilde{f}_A(t) - \tilde{f}_B(t))], \tag{86}$$

where $\tilde{f}_A(t)$ and $\tilde{f}_B(t)$ are the drift coefficients of $A_t$ and $B_t$, respectively. Decomposing gives us

$$\frac{\mathrm{d}}{\mathrm{d}t}\delta_t = \underbrace{-2\mathbb{E}[(A_t - B_t)(f(A_t, T - t) - f(B_t, T - t))]}_{C_1} \tag{87}$$

$$+ \underbrace{2\mathbb{E}[(A_t - B_t)g^2(T - t)(\nabla \log p_{T-t}(A_t) - s_\theta(B_t, T - t))]}_{C_2}. \tag{88}$$

By Assumption 3, we have

$$C_1 \leq -2\rho(T - t)\delta_t. \tag{89}$$

Next, we again decompose $C_2$ to get

$$C_2 = 2g^2(T - t)(\mathbb{E}[(A_t - B_t)](\nabla \log p_{T-t}(A_t) - \nabla \log p_{T-t}(B_t))$$
$$+ \mathbb{E}[(A_t - B_t)](\nabla \log p_{T-t}(B_t) - s_\theta(B_t, T - t))). \tag{90}$$

By Young's inequality and Assumptions 2 and 3, we obtain

$$C_2 \leq 2g^2(T - t)\left(L\delta_t + \lambda\delta_t + \frac{\epsilon_{\text{score}}^2}{4\lambda}\right) \tag{91}$$

for some hyperparameter $\lambda$. Therefore,

$$\frac{\mathrm{d}}{\mathrm{d}t}\delta_t \leq [-2\rho(T - t) + 2g^2(T - t)(L + \lambda)]\delta_t + \frac{\epsilon_{\text{score}}^2}{2\lambda}g^2(T - t). \tag{92}$$

Then we can define

$$I(t) := \int_{T-t}^T [-2\rho(s) + 2g^2(s)(L + \lambda)]\mathrm{d}s, \tag{93}$$

so when we apply Grönwall's inequality, we have

$$\delta_T \leq e^{I(T)}\delta_0 + \frac{\epsilon_{\text{score}}^2}{2\lambda}\int_0^T g^2(t)e^{I(T)-I(T-t)}\mathrm{d}t. \tag{94}$$

Finally, we get

$$\mathcal{W}_2(p_0, \text{Law}(Y_T)) \leq \sqrt{\mathcal{W}_2^2(p_T, p_{\text{noise}})e^{I(T)} + \frac{\epsilon_{\text{score}}^2}{2\lambda}\int_0^T g^2(t)e^{I(T)-I(T-t)}\mathrm{d}t}. \tag{95}$$

We again can apply the Wasserstein-2 distance to our setup. In particular,

$$I(t) = \int_{T-t}^T [-2\rho(s) + 2g^2(s)(L + \lambda)]\mathrm{d}s \tag{96}$$

$$= \int_{T-t}^T [\beta_s + 2(L + \lambda)\beta_s]\mathrm{d}s \tag{97}$$

$$= (1 + 2(L + \lambda))\int_{T-t}^T \beta_s \mathrm{d}s \tag{98}$$

$$= (1 + 2(L + \lambda))(\alpha_T - \alpha_{T-t}). \tag{99}$$

Thus, $I(T) = (1 + 2(L + \lambda))\alpha_T$. Additionally,

$$\mathcal{W}_2^2(p_T, p_{\text{noise}}) = \mathcal{W}_2^2(\mathcal{N}(m_T x_0, v_T I_d), \mathcal{N}(0, I)) \leq m_T^2 \mathbb{E}[\|x_0\|^2] + d(\sqrt{v_T} - 1)^2. \tag{100}$$

Since $m_T = \exp(-\frac{1}{2}\alpha_T)$ and $v_T = 1 - \exp(-\alpha_T)$, we have

$$\mathcal{W}_2^2(\mathcal{N}(m_T x_0, v_T I_d), \mathcal{N}(0, I)) = \exp(-\alpha_T)\|x_0\|^2 + d\left(1 - \sqrt{1 - \exp(-\alpha_T)}\right)^2. \tag{101}$$

We conclude

$$\mathcal{W}_2^2(p_0, \text{Law}(Y_T)) \leq (e^{-\alpha_T}\mathbb{E}[\|x_0\|^2] + d(1 - \sqrt{1 - \exp(-\alpha_T)})^2)(e^{(1+2(L+\lambda))\alpha_T})$$
$$+ \frac{\epsilon_{\text{score}}^2}{2\lambda}\int_0^T \beta_t e^{(1+2(L+\lambda))\alpha_t}\mathrm{d}t. \tag{102}$$

**Remark 5.** *Similar to Kwon et al. (2022), we assume an $L^\infty$ bound on score matching, and if we were to assume instead an $L^2$ bound, the result still holds as long as the score regularity in Assumption 3 is applied to the learned score instead of the Stein score function. For an $L^2$ bound on the score matching, see Gao et al. (2025).*

