# OpenReview forum: "Generative Diffusion Models for High-Dimensional Time Series"
_ICLR.cc/2026/Conference — Submitted to ICLR 2026_

### Official Review · Reviewer_HKjt · 2025-10-16

**Soundness:** 1
**Presentation:** 1
**Contribution:** 2
**Rating:** 2
**Confidence:** 5

**Summary:**

This paper focuses on the problem of time series modeling. Specifically, the proposed approach can be viewed as a combination of non-parametric estimation (i.e., Nadaraya–Watson estimation) and diffusion models. As I understand it, the generation process proceeds sequentially over time steps, where at each step a $\Delta$ value is computed and added iteratively to the previous point. This delta consists of two parts: a fixed component (e.g., the mean) and a residual component. The authors delegate the generation of the residual part to the diffusion model, while the fixed component is pre-determined through non-parametric estimation. In the experimental section, the authors evaluate the effectiveness of the proposed method on synthetic datasets.

**Strengths:**

The proposed decomposition-based time series generation framework grounded in non-parametric estimation is distinct from existing approaches such as multi-scale or trend–cycle decompositions. From my perspective, this design represents a interesting direction.

**Weaknesses:**

Overall, my reading experience with this paper was not very positive. I summarize several major weaknesses as follows:
1. **Unclear problem formulation and motivation.**
The paper lacks a precise problem definition or clear motivation. The direct transition from the *Introduction* to the *Method* section leaves readers puzzled—what exactly is this paper trying to achieve? Moreover, it remains unclear why the authors chose to combine non-parametric estimation with diffusion models. Why not simply use a diffusion model to jointly fit all components? I strongly recommend that the authors clearly articulate the problem statement and the motivation for their design, either in the *Introduction* or in a dedicated section.

2. **Poor writing and presentation quality.**
To illustrate with examples: in the *Introduction*, the authors briefly introduce diffusion models, followed by a very limited review of related work that is neither comprehensive nor contextualized. The contribution part then appears abruptly, with no conceptual transition.
In the *Method* and *Theoretical Analysis* sections, I strongly suggest including a table summarizing all mathematical symbols (e.g., (x), (X), (\hat{X}), etc.) and reorganizing the overall structure for better readability.
In the *Experiments* section, the authors seem to confuse the distinction between experiment setup and experiment results. Excessive attention is given to describing model parameters, while the analysis of results is extremely brief.

3. **The proposed method is rather trivial.**
In essence, the approach appears to be a straightforward combination of existing methods, without any meaningful mechanism to make them complement each other (i.e., achieving synergy beyond simple stacking). Moreover, the method has inherent limitations: it cannot generate long sequences and requires a known starting point for data augmentation. From a higher-level perspective, this is not substantially different from a classical autoregressive model.
Combined with the high inference cost of diffusion models, the proposed framework inherits the inefficiencies of both autoregressive and score-based models.
As for the theoretical analysis, while I did not examine every detail, it seems that the main conclusions are largely derived from existing results.
Additionally, *Algorithm 1* in the paper appears to be a pre-processing and joint training–generation framework rather than a “training algorithm” per se; I suggest renaming it accordingly.

4. **Experiments are too limited.**
The experimental evaluation is overly simplistic—there are no baselines, no real-world datasets, and no standard metrics such as KL divergence or FID. The authors themselves seem aware of these limitations, as the *Discussion* section lists many future directions, including comparisons with time-series diffusion models and sensitivity analyses.
Overall, the current level of empirical evaluation falls far short of what is expected for a top-tier venue like ICLR.

5. **Concerns about reproducibility.**
The paper does not release any source code or detailed implementation information, making it difficult for readers to reproduce or verify the results.

**Questions:**

See the Weaknesses part.

---

### Official Review · Reviewer_6j5L · 2025-10-31

**Soundness:** 2
**Presentation:** 1
**Contribution:** 1
**Rating:** 2
**Confidence:** 3

**Summary:**

The paper proposes generating time series by decomposing it into first and second moments and then fitting a score-based diffusion model. The authors provide some theoretical results. There is a simple synthetic experiment showing how it works.

**Strengths:**

The paper is decent to read. The method is somewhat sound and the theoretical results are good, if correct, I didn't check in detail. The experiment shows a good results.

**Weaknesses:**

There are no real experiments, only a single synthetic experiment. There are no baselines, only comparison to an oracle. This is not good enough for a machine learning conference, especially since there are so many different time series datasets that this can be applied to and so many models to compare to.

Formatting is bad. Algorithm 1 is too big, it should be 2 algorithms at least, one for sampling and one for training. Theoretical results don't seem that important to be in the main text. Figure 1 is low quality and takes too much space.

**Questions:**

No particular questions. Please see the weaknesses for my comments. If there were actual experiments I would ask about the runtime comparison, learning efficiency, choice of hyperparameters etc.

---

### Official Review · Reviewer_8yh6 · 2025-11-01

**Soundness:** 2
**Presentation:** 2
**Contribution:** 2
**Rating:** 4
**Confidence:** 3

**Summary:**

This paper introduces a two-stage pipeline for generating high-dimensional time series. The proposed method first employs a Nadaraya-Watson kernel estimator to decompose time series increments into conditional mean, covariance, and residuals. In the second stage, a score-based diffusion model is trained on these extracted residuals. The authors present a theoretical analysis providing finite-time convergence bounds in Total Variation (TV) and Wasserstein-2 (W2) metrics, and validate their approach on a synthetic 20-dimensional vector autoregressive process.

**Strengths:**

The primary strength of the paper lies in its elegant and intuitive approach. Decoupling the learning of deterministic dynamics from the modeling of the innovation distribution could potentially simplify the task for the score network, a promising direction for time-series generation. Furthermore, the effort to provide a rigorous theoretical analysis and to use quantitative diagnostics (such as empirical TV/W2 distances and test functionals) for evaluation, rather than relying solely on qualitative plots, is commendable in principle.

**Weaknesses:**

1. Incomplete convergence guarantees. The reverse-time bounds are derived under the true residual distribution, but training uses residuals estimated via kernel smoothing. The stage-one estimation errors in $\hat{\mu}$ and $\hat{a}$ never enter the analysis, so the theoretical guarantees do not reflect the actual pipeline.

2. Unsubstantiated regularity assumptions. Assumption 3 requires the true score $\nabla \log p_t(x)$ to be globally Lipschitz. The experiments use Gaussian-mixture residuals, yet the paper does not provide theoretical or empirical justification that these conditions are satisfied, leaving the applicability of Theorem 1 uncertain.

3. Implementation opacity of the kernel stage. Although a data-adaptive $k$-nearest-neighbour bandwidth is mentioned, the paper gives no guidance on how this scales in 20 dimensions nor on keeping the estimated covariance matrices positive definite. Reproducibility remains in doubt.

4. Limited and contradictory empirical evidence. The evaluation considers only one synthetic process without baselines or ablations, and Table 1 reports a ≈40 % bias on the basket functional despite tiny Monte Carlo errors, directly undermining the claim that the model preserves key statistics.

**Questions:**

1. How are the kernel bandwidths selected in 20 dimensions? What measures are taken to ensure the estimated covariance matrices remain positive definite and well-conditioned, and what is the computational complexity of this stage?

2. Can you provide a rigorous verification that your chosen OU process satisfies the one-sided Lipschitz condition in Assumption 3? More critically, can you provide theoretical justification or empirical evidence that the score function of a Gaussian Mixture Model, after undergoing the OU diffusion process, remains globally Lipschitz continuous across all time steps $t \in [0,T]$? Without such justification, how do you defend the validity of Theorem 1?

3. How would the theoretical analysis and the final convergence bounds change if the estimation error from the first stage (i.e., errors in $\hat{\mu}$ and $\hat{a}$) were to be properly propagated through to the reverse SDE?

4. How do you explain the large discrepancy between the oracle and model-generated functional estimates in Table 1? Does this significant bias fall within your theoretical error bounds, and what does it imply about the model's ability to capture the true data distribution?

---

### Official Review · Reviewer_SmmW · 2025-11-02

**Soundness:** 3
**Presentation:** 2
**Contribution:** 2
**Rating:** 4
**Confidence:** 2

**Summary:**

The paper proposes a two-stage pipeline for multivariate time-series generation: first, estimate conditional mean and covariance via Nadaraya–Watson kernel regression to extract residuals; second, train a score-based diffusion (variance-preserving OU/DDPM) on those residuals.

**Strengths:**

- Clear, modular algorithm (nonparametric residualization + diffusion), with an explicit training-and-sampling procedure
- Theoretical bounds expose how noise schedules impact convergence; W2 analysis via coupled SDEs is carefully laid out
- Synthetic study includes interpretable probes (functionals) and visual checks of residual geometry, including multimodality

**Weaknesses:**

- **Metric-dependent behavior and ranking variance.** Figure 1 suggests markedly different trends: W2 steadily decreases while TV appears comparatively flat, implying that relative model quality could look very different depending on the metric used. This raises concerns about stability of conclusions under metric choice and whether improvements are universal or metric-specific.
- **Limited datasets and scope.** Evaluation is confined to a single synthetic setting (vector AR(1) with mixture innovations) without real-world time-series or domain-standard baselines (e.g., time-series DDPMs, latent-SDEs) to contextualize gains. The Discussion itself calls for comparisons and scaling studies, underscoring the current evaluation gap.
- **Lack of analysis on why it works.** While empirical distances and functionals are reported, there is little ablation or diagnostic analysis explaining *why* kernel residualization plus diffusion yields the observed behavior (e.g., how bandwidth choice, lag augmentation, or residual non-Gaussianity affect learning and bounds). The paper notes future stress tests but does not include them.
- **Strong assumptions and practicality.** The convergence results rely on regularity conditions (e.g., Lipschitz scores, drift growth) that may be hard to verify for high-dimensional, non-Markov real data. It is unclear how violations (e.g., heavy tails, regime shifts) would affect guarantees or practice.
- **Algorithmic sensitivity not quantified.** The first stage uses locally adaptive k-NN bandwidths; there is no sensitivity study for bandwidth, lag selection, or Cholesky factor ambiguity (rotation) that could materially change residual structure and thus the diffusion target.

**Questions:**

See the weaknesses above.

---

### Meta-Review · Area_Chair_4Jud · 2025-12-15

**Summary:**

Across four reviews, the paper received consistently low scores (two explicit rejects and two borderline rejects), with strong agreement on several major shortcomings that motivate rejection.

Primary concerns center on the lack of sufficient empirical validation. All reviewers note that the evaluation is limited to a single synthetic AR(1) process with Gaussian-mixture noise, with no real-world datasets, no comparisons to standard time-series diffusion baselines, and no ablations that meaningfully justify the design choices. This is considered far below the empirical standard expected for ICLR, particularly for a generative modeling paper. Reviewers 6j5L and HKjt explicitly state that such an evaluation alone is not acceptable for the venue.

A second major concern is the disconnect between theory and practice. Reviewer 8yh6 highlights that the convergence analysis applies to an idealized setting with access to true residuals, while the actual method relies on kernel-estimated residuals. Although the paper later introduces an abstract Assumption 4 to account for kernel error, this assumption is not quantified, empirically validated, or propagated in a way that makes the guarantees practically meaningful. As a result, the theoretical results do not convincingly apply to the implemented algorithm.

Third, reviewers raise strong concerns about assumptions and regularity conditions. The global Lipschitz score assumption, central to the Wasserstein bounds, is viewed as unrealistic outside the carefully constructed synthetic setting. While the appendix provides a proof for Gaussian-mixture residuals diffused by an OU process, reviewers note that this does not generalize to realistic, high-dimensional time-series data with heavy tails or regime changes. This limits the broader relevance of the theoretical contribution.

Fourth, the methodological novelty is viewed as limited. Multiple reviewers characterize the approach as a relatively straightforward combination of kernel regression and diffusion models, without demonstrating clear synergy or advantages over directly applying conditional diffusion or existing time-series diffusion models. Reviewer HKjt emphasizes that, at a high level, the approach resembles a classical autoregressive model augmented with an expensive diffusion-based noise generator, inheriting inefficiencies from both.

Finally, presentation and clarity issues further weaken the submission. Reviewers point out poor organization, lack of a clear problem statement and motivation, confusing notation, oversized algorithms, low-quality figures, and insufficient implementation details. These issues significantly hinder readability and reproducibility.

Overall, while reviewers acknowledge that the idea of residualizing dynamics before diffusion is intuitively appealing, the paper does not meet ICLR standards in empirical rigor, practical relevance, or clarity, and the theoretical contribution is insufficiently connected to the actual method.

**Reviewer Concerns:**

Partially addressed concerns:
	•	The authors provided additional clarification in the appendix regarding the Lipschitz score assumption for Gaussian-mixture residuals under an OU diffusion, which addresses one specific theoretical question raised by Reviewer 8yh6. However, this clarification remains limited to the synthetic setting and does not alleviate concerns about applicability to real data.
	•	The paper includes some sensitivity analyses (lag length, bandwidth choice, Cholesky rotation) reported in the experiments section and tables. These partially respond to concerns about kernel-stage sensitivity raised by Reviewers SmmW and 8yh6.

Outstanding concerns (not resolved):
	•	Lack of real-world experiments and baselines remains entirely unaddressed. No new datasets, comparisons, or benchmarks are provided.
	•	The theory–practice gap persists: kernel estimation error is still abstracted into an unverified assumption, and the guarantees do not convincingly apply to the implemented pipeline.
	•	Metric-dependent empirical behavior (TV vs. W2 divergence) is acknowledged but not resolved; conclusions remain sensitive to the choice of metric.
	•	Large functional bias in Table 3 (e.g., ≈40% error on the basket functional) remains unexplained and undermines claims that key statistics are preserved.
	•	Methodological novelty and motivation concerns remain: the rebuttal does not convincingly explain why this two-stage approach is preferable to existing time-series diffusion or latent-SDE methods.
	•	Presentation and reproducibility issues (unclear structure, formatting, missing implementation details, no code release) are not substantially improved.

In summary, while the rebuttal clarifies some narrow technical points, it does not address the core reasons for rejection, particularly the lack of compelling empirical evidence and the limited practical impact of the theoretical results.

**Reviewer Scores:**

Based on the reviews, rebuttal, and limited scope of addressed concerns, I do not expect any reviewer to substantially increase their score had full discussion occurred:
	•	Reviewer SmmW (Score: 4, low confidence)
Likely to remain at 4 or potentially decrease to 3, given the acknowledgment of limited expertise and the persistence of evaluation and practicality concerns.
	•	Reviewer 8yh6 (Score: 4)
Likely to remain at 4. While some theoretical clarifications were provided, the core critique regarding incomplete guarantees and insufficient experiments remains unresolved.
	•	Reviewer 6j5L (Score: 2)
Likely to remain at 2. The absence of real experiments and baselines—the primary concern—was not addressed.
	•	Reviewer HKjt (Score: 2, very high confidence)
Very unlikely to change. The reviewer expressed strong concerns about novelty, motivation, empirical adequacy, and presentation, none of which were meaningfully resolved.

Thus, the overall assessment would remain clearly below the acceptance threshold.

---

### Decision · Program_Chairs · 2026-01-26

Reject